# Curcumin Metabolite Tetrahydrocurcumin in the Treatment of Eye Diseases

**DOI:** 10.3390/ijms22010212

**Published:** 2020-12-28

**Authors:** Yu-Wen Kao, Sheng-Kai Hsu, Jeff Yi-Fu Chen, I-Ling Lin, Kuo-Jen Chen, Po-Yen Lee, Hui-Suan Ng, Chien-Chih Chiu, Kai-Chun Cheng

**Affiliations:** 1Department of Medicine, Kaohsiung Medical University, Kaohsiung 807, Taiwan; u106001068@kmu.edu.tw; 2Department of Biotechnology, Kaohsiung Medical University, Kaohsiung 807, Taiwan; b043100050@gmail.com (S.-K.H.); yifuc@kmu.edu.tw (J.Y.-F.C.); 3Department of Medical Laboratory Science and Biotechnology, Kaohsiung Medical University, Kaohsiung 807, Taiwan; linili@kmu.edu.tw; 4Department of Ophthalmology, Kaohsiung Municipal Siaogang Hospital, Kaohsiung 812, Taiwan; 0870649@kmhk.org.tw; 5Department of Ophthalmology, Kaohsiung Medical University Hospital, Kaohsiung 807, Taiwan; maco69@gmail.com; 6Faculty of Applied Sciences, UCSI University, UCSI Heights, Cheras 56000, Kuala Lumpur, Malaysia; GrraceNg@ucsiuniversity.edu.my; 7Department of Biological Sciences, National Sun Yat-sen University, Kaohsiung 804, Taiwan; 8Center for Cancer Research, Kaohsiung Medical University, Kaohsiung 807, Taiwan; 9Department of Medical Research, Kaohsiung Medical University Hospital, Kaohsiung 807, Taiwan; 10The Graduate Institute of Medicine, Kaohsiung Medical University, Kaohsiung 807, Taiwan; 11Department of Ophthalmology, School of Medicine, College of Medicine, Kaohsiung Medical University, Kaohsiung 807, Taiwan

**Keywords:** curcumin, COX, SIRT1, tetrahydrocurcumin (THC), ophthalmic diseases, VEGF

## Abstract

Curcumin is one of the most valuable natural products due to its pharmacological activities. However, the low bioavailability of curcumin has long been a problem for its medicinal use. Large studies have been conducted to improve the use of curcumin; among these studies, curcumin metabolites have become a relatively new research focus over the past few years. Additionally, accumulating evidence suggests that curcumin or curcuminoid metabolites have similar or better biological activity than the precursor of curcumin. Recent studies focus on the protective role of plasma tetrahydrocurcumin (THC), a main metabolite of curcumin, against tumors and chronic inflammatory diseases. Nevertheless, studies of THC in eye diseases have not yet been conducted. Since ophthalmic conditions play a crucial role in worldwide public health, the prevention and treatment of ophthalmic diseases are of great concern. Therefore, the present study investigated the antioxidative, anti-inflammatory, antiangiogenic, and neuroprotective effects of THC on four major ocular diseases: age-related cataracts, glaucoma, age-related macular degeneration (AMD), and diabetic retinopathy (DR). While this study aimed to show curcumin as a promising potential solution for eye conditions and discusses the involved mechanistic pathways, further work is required for the clinical application of curcumin.

## 1. Introduction

Due to lifestyle changes and increased longevity, increased attention is being paid to the population with visual impairment worldwide [1,2]. For this purpose, researchers are dedicated to finding specific botanical compounds, especially those from curcumin, widely used throughout history, to intervene in common mechanisms of damage in ocular pathologies [3]. Curcumin is a major polyphenol from *Curcuma longa* (*Zingiberaceae)*, which has a long history as a spice and folk remedy in China and India. In Ayurvedic medicine, curcumin is used to treat eye infections and other diseases. Early Europeans introduced curcumin from Asia to the Western world, and Western medicine practitioners have discovered a surprisingly wide range of beneficial properties of this ancient remedy [4].

Considering these facts, we decided to investigate the effects of pharmaceutical products from curcumin on eye disorders and determine the ophthalmic diseases that can benefit from these products.

According to statistics from the World Health Organization (WHO), the total number of individuals of all ages with visual impairment worldwide in 2010 was estimated to have been 285 million, and among these individuals, 39 million were blind. Visual impairment and blindness have become major global health issues, and among the causes of visual impairment, preventable causes account for 80% of the total global burden.

The occurrence of visual impairment and blindness in developed countries is lower than that in developing regions, such as sub-Saharan Africa and South Asia [5,6]. Due to the increasing age of the world’s population and changes in the age distribution worldwide, efforts to improve individuals’ quality of life with visual impairment and blindness and decrease global economic costs related to visual disorders are needed. Therefore, this study assessed the pharmaceutical pathways of curcumin metabolites to eliminate the burden of unnecessary blindness and vision impairment.

## 2. Curcumin

### 2.1. Curcumin: Limitations

As a common remedy, curcumin possesses diverse properties, such as its anti-inflammatory and antioxidant capacity. Several studies demonstrated that curcumin could be a wound-healing agent when topically administered. It exerts benefits during the inflammation, proliferation, and remodeling phases in the wound healing process [7]. However, when considering the effects of systemic absorption, access to curcumin’s pharmacological application is limited due to its poor solubility, low gastrointestinal absorption, and fast hepatic and intestinal metabolism [8,9]. Therefore, modifying curcumin bioavailability is the most important step to promote its beneficial effects against several ocular diseases [9,10]. In terms of improving the bioavailability of curcumin, in the next paragraphs, we will discuss the following approaches to modify curcumin: delivery formulations and metabolites.

### 2.2. Modulation of Curcumin: Delivery Formulations

Curcumin discovery dates back approximately two centuries when curcumin was discovered from the rhizomes of *Curcuma longa* of the ginger family [11]. Unfortunately, the hydrophobic polyphenol structure of curcumin significantly decreases its bioavailability. Briefly, there are three barriers to curcumin’s therapeutic potential: its low solubility, low absorption ratio, and fast metabolic rate. Previous studies have suggested the use of micelles, liposomes, phospholipid complexes, microemulsions, nanoemulsions, and several nanostructured carriers as delivery systems for curcumin [12]. First, the hydrophobic curcumin loaded into the core of copolymer micelles could be easy to reconstitute in water. Next, liposomes can also carry hydrophobic curcumin in their phospholipid bilayer vesicles. Finally, nanoemulsions not only have a hydrophobic liquid core but are also stabilized by a surfactant monolayer, which effectively reduces the interfacial tension of the droplets. As can be seen, size and surface properties are critical for the cellular uptake of a substance. Four broad formulation strategies will be discussed below, which have been used to enhance curcumin bioavailability: lipid addition, absorption and dispersion on matrices, particle size reduction, and surface property modulation.

#### 2.2.1. Lipid Addition

Early approaches combined existing agents, such as piperine and turmeric oil. Piperine is a major bioactive pepper component that is rapidly absorbed through the gastrointestinal (GI) tract and does not undergo metabolic changes during its absorption from the intestine. The maximum plasma concentration of piperine is attained at approximately 6 h. In 1998, Shoba et al. showed that the presence of piperine, an inhibitor of hepatic and intestinal glucuronidation, significantly improved the curcumin plasma concentration, the extent of curcumin absorption, and the bioavailability of curcumin in both a rat model and humans [13]. Apart from piperine, the addition of lipids to curcumin is another option. The reconstitution of curcumin with turmeric’s noncurcuminoid components had a synergistic effect and substantially increased the efficacy and bioavailability of cumin, with the resulting patented formulation trademarked as BCM-95^®^. The relative bioavailability of BCM-95^®^ was shown to be approximately 6.93- and 6.3-fold that of normal curcumin and curcumin–lecithin–piperine, respectively. Consequently, BCM-95^®^ has extensive antioxidative applications in various diseases [14].

#### 2.2.2. Absorption and Dispersion on Matrices

Newer formulations have involved the dispersion of curcumin onto various matrices. For instance, a drug with the trade name Meriva^®^ uses a novel phytosome structure to enhance curcumin’s capacity to cross lipid membranes and reach the systemic circulation. Meriva^®^, the microcrystalline cellulose structure combined with soy lecithin phosphatidylcholine, reaches a therapeutic level in the eye when administered at standard dosage. [15] Therefore, the success of this approach implies its significant potential for effective ophthalmic drug therapy.

#### 2.2.3. Particle Size Reduction

The most common curcumin formulation uses various techniques to minimize its particle size. Among the resulting pharmaceutical products, nanocrystal conjugates are the most effective [10]. For example, nanocrystal curcumin can be encapsulated in polyethylene glycol (PEG) and/or poly lactic-co-glycolic acid (PLGA), which acts as a carrier for oral delivery of curcumin; the formulation of PLGA-PEG blended nanoparticles increased curcumin bioavailability by over 55-fold [16]. In another study, Gangwar et al. experimented with the conjugation and loading of curcumin with silica nanoparticles to improve its aqueous solubility [17]. Compared with free curcumin, due to its nanoscale size, this formulation of curcumin had a faster dissolution rate, faster cellular uptake in vitro, and improved solubility and stability [18,19].

#### 2.2.4. Surface Property Modulation

Other critical factors concerning curcumin bioabsorption are its surface properties. Enhancing curcumin’s surface charge and adhesion properties could improve problematic low gastrointestinal absorption by enhancing its contact with the intestinal mucosal epithelium. Since cell membranes are negatively charged, nanocrystal curcumin’s slightly positive surface charge may increase its interaction with epithelium.

Research concerning the ophthalmic use of curcumin with altered physicochemical properties within the formulation mentioned above has been conducted, three examples of which follow. First, the preparation of a nanoparticle formulation of curcumin consisting of a thermosensitive ophthalmic nanogel, CUR-CNLC-GEL (a nanogel containing cationic nanostructured lipid carriers), was developed. CUR-CNLC-GEL was confirmed to enhance the corneal permeation and retention capacity of curcumin and increase bioavailability in the aqueous humor in vivo and in vitro [20]. Furthermore, the administration of curcumin incorporated into albumin nanoparticles (Cur-BSA-NPs-Gel) to rabbits in an ophthalmic experiment showed that Cur-BSA-NPs-Gel is considered safe for ophthalmic drug delivery. The Cur-BSA-NPs-Gel formulation significantly increased the effect of curcumin in the aqueous humor [21]. Finally, the role of curcumin-encapsulated methoxypoly(ethylene glycol)-poly(caprolactone) (MePEG-PCL) nanoparticles in the prevention of corneal neovascularization was successfully characterized. Compared with curcumin, MePEG-PCL nanoparticles more significantly suppressed vascular endothelial growth factor (VEGF), inflammatory cytokines, and matrix metalloproteinases (MMPs) to prevent angiogenic sprouting in vitro [22].

### 2.3. Modulation of Curcumin: Curcumin Metabolites

The solutions, as mentioned above, enhance the bioavailability of curcumin. However, despite its low absorption, curcumin still possesses significant biological effects. Therefore, scientists recently focused on exploring curcumin metabolites, hoping to discover new methods to increase curcumin’s therapeutic potential (Figure 1).

Curcumin is mainly metabolized into dihydrocurcumin (DHC), tetrahydrocurcumin (THC), hexahydrocurcumin (HHC), and octahydrocurcumin (OHC), the final form of hydrogenated curcumin. Among the curcumin metabolites, THC–glucuronoside, a conjugated form of curcumin, exhibited the greatest biliary concentration in rats [23,24]. THC was also shown to be the primary metabolite responsible for curcumin’s various biological properties. Compared with other curcumin metabolites, THC, the major plasma metabolite of curcumin, demonstrates higher solubility at physiological pH; a longer half-life in plasma at 37 °C, and higher antioxidant, anti-inflammatory and anticancer activities. Experiments have been conducted to explore the antioxidant and anti-inflammatory effects of THC and OHC [25]. Compared with curcumin, THC and OHC are more effective in suppressing nuclear factor-κB (NF-κB) and inhibiting the expression of cyclooxygenase 2 (COX-2). In Vitro evidence of these compounds’ antioxidative effects has been reported, but in vivo experiments are still being conducted [25]. Since the field of curcumin metabolites is rather novel, the pathways involved in the effects of THC and OHC in the eye have rarely been published. In this context, the present review addresses the possible mechanism through which THC and OHC interfere with the development of ocular diseases (Figure 2).

## 3. Therapeutic Potential of Curcumin in Ophthalmology

Curcumin has been shown to have considerable potential health benefits in recent studies. A search yielded almost twenty thousand manuscripts on this topic from 2014 to 2019, most of which are related to the use of curcumin derivatives against cancer and cardiovascular diseases [26]. Nevertheless, research concerning the therapeutic potential of curcumin metabolites in ophthalmology is rather limited. Since combating avoidable visual impairment and blindness is important in public health policies throughout the world, the development of THC, a curcumin metabolite, may show promise in ophthalmology.

The following review discusses the effects of curcumin in eye conditions. The first section lists the main ophthalmic conditions that affect modern society and the pathological mechanisms of these visual impairment forms. The second section discusses THC pathways, which may be beneficial in the treatment of visual impairments.

### 3.1. Main Ophthalmic Conditions in Modern Society

Aging affects all the eye structures, triggering various eye conditions; consequently, the prevalence of blindness and moderate to severe visual impairment (MSVI) is much greater in elderly individuals [27]. According to statistics reported in 2018, the global population of individuals with blindness in 2015 was approximately 36 million, and slightly more than 216 million individuals had MSVI. At all ages, the leading cause of blindness is cataracts, followed by refractive error, glaucoma, age-related macular degeneration (AMD), and corneal opacity. The causes of MSVI rank as follows: refractive error, cataracts, AMD, glaucoma, and diabetic retinopathy (DR) [28,29]. Therefore, cataracts, glaucoma, AMD, and DR account for the largest percentage of serious eye disease cases.

Before discussing the ophthalmic conditions’ mechanisms, some risk factors involved in this public health phenomenon are discussed.

First, the most significant underlying intrinsic factors of cataracts are age and sex (female). Age is crucial for the development of cataracts due to accumulated oxidative stress over time. According to the National Eye Institute (NEI) and its statistics, updated in 2019, the increased risk of cataracts starts around age 40. Poor nutrition and smoking are examples of extrinsic factors associated with cataracts [30]. A cataract is crucial in developing countries, where the majority of cataract cases result in blindness. Due to an aging population in those areas, the cataract incidence has increased and has gained increasing attention [31,32].

Second, risk factors for open-angle glaucoma are as follows. Ocular factors include increased intraocular pressure, ocular perfusion pressure, and optic disc hemorrhage. Systemic factors, such as systemic hypertension, type 2 diabetes mellitus, and lipid dysregulation may also increase glaucoma risks. Moreover, age, smoking, family history, and genetic factors could also be risk factors [33]. Although glaucoma may occur at any age, a relationship was found between open-angle glaucoma and increasing age due to other age-related diseases. The health deficits include vascular diseases, diabetes, and macular degeneration, which may occur with aging [34,35,36].

Third, as for AMD, smoking has the most significant relationship with both wet and dry AMD, aside from age. Other controllable risk factors are diet and cardiovascular health. Genetics and aging are also significantly related to AMD [37]. Statistics of the National Eye Institute suggest that AMD is most common among older white Caucasians, and the prevalence rate increases significantly over age 80.

Finally, DR is strongly associated with long diabetes duration and poor glycemic and blood pressure control. Therefore, apart from diabetes, hypertension, and obesity are most significantly associated with DR [38,39]. The aging global population and rising prevalence of obesity have resulted in the increased prevalence of diabetes and diabetic retinopathy. Besides, with the improvement in diabetes treatment, more patients with diabetes live long enough for DR to develop [40]. Generally, eye diseases are related to aging and several systemic conditions, such as diabetes mellitus and vascular diseases. Therefore, these ophthalmic conditions have a high impact on the global health burden.

In the section that follows, the causes and mechanisms of ophthalmic conditions mentioned above will be assessed in order.

#### 3.1.1. Age-Related Cataract

A cataract is any type of opacification of the crystalline lens in the eye. A decline in the lens’s optical quality, which is normally clear, can lead to visual symptoms. The main pathogenesis of age-related cataracts is the modification of lens proteins under oxidative stress. Severe modification of the proteins and their unusual interactions lead to inappropriate protein folding and aggregation, causing lens opacification. Accumulation of free radicals in the eye lens is a common initiating factor in cataract formation. The increase in free radicals induces oxidative stress, and lipid peroxidation (LPO) and the aggregation of malfunctioning proteins result from oxidative damage. Babizhayev, Deyev, and Linberg injected LPO products into the vitreous after finding a correlation between the accumulation of a fluorescent end product of LPO and the degree of lens opacity. The injection of LPO products induced cataract, implying that the lens fiber’s peroxide-induced damage may be one of the important triggers that initiate cataractogenesis [41]. As an important structural protein in the lens, α-crystallin helps fold and stabilize other lens proteins. The molecular chaperone function of α-crystallin subunits is to prevent aggregation of proteins under stress conditions. α-Crystallin interacts with proteins that are about to precipitate [42]. The hydrophobic sites of both α-crystallin and partially unfolded proteins integrate; therefore, the aggregation-prone proteins are held in a refolding competent state [43]. However, after various stress factors, especially oxidative stress, deteriorate the chaperone-like function of α-crystallin, it cannot maintain lens transparency, potentially leading to cataract formation. Many of the structural proteins, especially α-crystallin, contain an abundance of -SH groups highly susceptible to oxidative damage. Redox reactions between SH-containing proteins and glutathione result in the accumulation of malfunctioning proteins and a decrease in reduced glutathione (GSH), which accelerates cataract formation [44,45].

In addition to oxidation, the glycation of lens proteins appears in various types of cataract. Glycation enhances protein unfolding and alters the physicochemical properties and functions of proteins [46]. αB-crystallin (HSPB5) is a chaperone responsible for the alleviation of unfolded proteins. However, excessive accumulation of unfolded proteins could lead to strong ER stress and apoptosis in retinal pigment epithelium (RPE) cells; therefore, αB-crystallin serves as a significant modulator of ER stress-induced cell death. Preliminary evidence has suggested that silencing of αB-crystallin via siRNA results in ER stress, subsequently leading to elevated ROS generation and reduced MnSOD activity; this causes cell damage to human RPE cells. Nevertheless, upregulation of αB-crystallin reversely prevents RPE cells from ER stress-induced apoptosis via inhibition of C/EBP homologous protein (CHOP) and caspase 3 [47].

ER stress is transduced via at least three signaling pathways: the IRE1α-dependent pathway, ATF6-dependent pathway, and PERK-dependent pathway (PERK, protein kinase RNA-like ER kinase). In a study by Berthoud et al., P-PERK immunostaining was significantly higher in mice with nuclear cataracts than in wild-type mice with normal lenses. C/EBP homologous protein (CHOP) transcripts associated with ATF4 levels are also increased in homozygous lenses, suggesting that activation of the PERK-dependent pathway is related to unfolded protein response (UPR) activation in the lens, leading to cataract [48].

Accordingly, the ideal method to correct age-related cataracts is the application of antioxidants. However, current studies have demonstrated that antioxidants have little effect after cataract formation. The only treatment to prevent the progression and development of cataracts is the surgical removal of the cloudy lens. Although medical treatments for cataracts have been administrated, models of antioxidant treatments, such as curcumin metabolites, which will be discussed later, may be applied to intervene before age-related cataract formation.

#### 3.1.2. Glaucoma

Glaucoma is a series of progressive optic neuropathies. The degeneration of retinal ganglion cells is mainly attributed to a surge in eye pressure after aqueous humor [49]. Other factors include chemical injury, inflammatory conditions, and changes in vessel density [50,51,52,53]. Therefore, neuroprotection through curcumin metabolism may be a method of preventing glaucoma [50,54]. Unfortunately, there is little evidence for the mechanisms by which these agents prevent glaucoma progression since the pathophysiological mechanisms of neural damage are not fully understood, and the clinical trials of these agents have not been conclusive. To date, the management of glaucoma is achieved by targeting intraocular pressure [55,56]. A broad collaborative effort to identify methods for neuroprotection against ophthalmic diseases is ongoing, with curcumin metabolites serving as the primary candidates.

#### 3.1.3. AMD

AMD and DR are two major causes of visual impairment due to changes in lifestyle and increased longevity [1]. AMD affects the retina’s macular region, causing progressive loss of vision in the center of the visual field. Changes in early-stage AMD include drusen and abnormalities of the retinal pigment epithelium (RPE), while late-stage AMD is divided into two types: neovascular (also known as the wet form of AMD) and non-neovascular (the dry form of AMD). Several pathways, such as choroidal ischemia and oxidative damage in RPE, have been implicated in AMD’s pathogenesis.

In recent years, therapeutic targets have focused on VEGF, a key regulator of vascular growth, and neovascular regression [57]. Several types of retinal cells, including the RPE, astrocytes, Müller cells, vascular endothelium, and ganglion cells, possess VEGF receptors [58,59,60]. In tissue under normoxia, retinal cells produce moderate VEGF to support existing blood vessels. However, hypoxia episodes, a crucial stimulus of VEGF gene expression, are related to vascularization development [61]. The activation of VEGFR-2 in cells in the RPE increases vascular permeability through the endothelial nitric oxide synthase (eNOS) pathway, promotes proliferation through the MEK/extracellular signal-regulated kinase (ERK) pathway, and provokes migration through the mitogen-activated protein kinase (MAPK) pathway. These processes result in angiogenesis, which plays an important role in the formation of AMD [62]. Under these circumstances, curcumin metabolites may play a role in AMD remission as antioxidants and VEGF inhibitors.

#### 3.1.4. DR

DR is a microvascular complication of diabetes mellitus. Early DR, also called nonproliferative diabetes retinopathy (NPDR), is characterized by weakened retina vessels. Microaneurysms protrude from vessel walls and leak fluid and blood into the retina. Advanced DR, known as proliferative diabetic retinopathy (PDR), is a complication resulting from new blood vessels’ irregular growth. Several pathogenic mechanisms are involved in DR, including VEGF, oxidative stress, ER stress, inflammation, and autophagy. Hyperglycemia leads to the dysfunction of the electron transport chain, causing the accumulation of reactive oxygen species (ROS) in mitochondria. It has been shown that DR occurs through chronic exposure to a high glucose level and the diacylglycerol-protein kinase C (DAG-PKC) molecular signaling pathway. Hyperglycemia promotes the synthesis and activity of DAG (diacylglycerol) and then triggers the PKC’s activation (protein kinase C) pathway. However, hyperglycemia-induced ROS can also induce the PKC pathway. Activated PKC can lead to several vascular abnormalities, such as increases in permeability and angiogenesis [63]. Furthermore, high glucose also activates the p38 MAPK signaling pathway, which initiates inflammation and subsequently induces apoptosis of endothelial cells and pericytes within retinal capillaries [64]. Kowluru et al. reported that diabetic-induced oxidative stresses could epigenetically result in the inactivation of MnSOD, an important enzyme in the removal of superoxide radicals, through elevated H4K20me3, acetyl H3K9, and p65 at the promoter of *sod2* (encoding MnSOD) [65]. This leads to DR due to the accumulation of reactive oxidative species and retinal capillary cell apoptosis [65,66]. Other research suggested that regardless of the type of diabetes, a high glucose condition can induce ROS and impair the balance of DNMT1 expression; however, curcumin can restore the activity and expression of DNMT1, which protects RPE from oxidative stress [67]. Along with ER stress and inflammation pathways, oxidative stress increases autophagy in the retina of diabetic patients. Autophagy acts as a double-edged sword in the modulation of several conditions in the body. Mild autophagic stress can lead to cell survival; however, severe dysregulated autophagy can initiate and deteriorate DR [68].

Additionally, oxidative stress due to hyperglycemia promotes macrophage migration and foam cell formation. Macrophages and foam cells release growth factors, resulting in plaque formation in the retina. THC, the main metabolite of curcumin, may modify impaired platelet function and coagulation abnormalities, which will be discussed in a later paragraph [69] (Table 1).

### 3.2. Therapeutic Potential of the THC Pathway

This paragraph follows from the previous chapter, which outlined curcumin and its major plasma metabolite, THC, and its mechanisms in the ocular diseases mentioned above. THC is not naturally found in turmeric extract powders but is found in plasma after curcuminoid ingestion. THC is the focus of the present study because it is a major metabolite of curcumin and exhibited activities similar to those of curcumin. In contrast, other identified metabolites, including the conjugates curcumin glucuronide and curcumin sulfate, are less biologically active than curcumin [71]. Moreover, compared with curcumin, THC is more stable and has a longer degradation half-life in buffers at various pH values and plasma [23]. In conclusion, THC has several potential protective benefits for the human body and has thus been the focus of recent studies (Figure 3).

### 3.3. Effect of THC on Antioxidative Stress

The eye is constantly exposed to all types of oxidative stress. ROS are the product of many mitochondria mechanisms and play a vital role in eye pathogenesis. ROS generation is often stimulated by the accumulation of advanced glycation end products (AGEs) in the lens in age-related cataract patients. Moreover, preliminary evidence demonstrated that hyperglycemia and increased ROS could result in downregulation of nicotinamide adenine dinucleotide phosphate (NADPH) and upregulation of NADPH oxidase (NOX) via phosphorylation of NOX. NADPH serves as a ROS scavenger because of glutathione regeneration; NOX is an enzyme that primarily generates different ROS (e.g., superoxide and hydrogen peroxide), which causes a vicious cycle of elevated ROS levels [64,72]. An experiment conducted by Suryanarayana et al. suggested that among the concentrations tested, 0.002% curcumin had the greatest antioxidant and antiglycation effects. Curcumin at a 0.002% concentration inhibited AGE fluorescence in the lens. This delayed the onset and maturation of age-related cataract [73].

Superoxide dismutase (SOD) and glutathione peroxidase are two of the main superoxide-scavenging systems in the cell. SOD-1 knockout in mice was observed to increase the risk of macular degeneration development [74]. Mice with defective SOD-2 also showed progressive retinal thinning and changes within the photoreceptor layer [75]. The enzymatic activities of SOD and glutathione peroxidase, ROS scavengers, and levels of the oxidative stress indicator malondialdehyde (MSA) were substantially reversed after THC treatment in a previous examination [76]. Significant increases in SOD and glutathione peroxidase strongly imply the antioxidative capacity of THC.

A decline in sirtuin-1 (SIRT1) was related to SOD’s reduced levels of SOD since SIRT1 deacetylates SOD [77]. SIRT1 is an NAD-dependent enzyme that deacetylates various substrates, contributing to a range of cellular regulatory mechanisms, such as gene expression, metabolism, and aging. The most important function of SIRT1 is its alleviation of inflammation by inhibiting NF-κB signaling and suppressing oxidative stress. Previous studies have demonstrated the dysfunction of SIRT1 in ocular diseases, and the knockdown of SIRT1 was associated with cataract, glaucoma, AMD, and DR [78]. In contrast, ectopic upregulation of SIRT1 served to protect against oxidative stress-induced impairments in several eye tissues, including the RPE, cornea, and lens. SIRT1 played a role in oxidative stress and was shown to have a significant neuroprotective effect in mice with an optic nerve crush injury [79]. Li et al. examined the correlation between THC and SIRT1 in diabetic cardiomyopathy and discovered that THC administration ameliorates oxidative stress by activating SIRT1. Suppression of the ROS-stimulated TGB-β-1 pathway against the decapentaplegic homolog 3 (Smad3) fibrotic pathway was also promoted by THC treatment [76]. This study provided insight into THC’s potential protective mechanism by activating SIRT1 and thus decreasing ROS, and the effectiveness of THC in ameliorating fibrosis.

The above is a brief review of the antioxidative function of THC through the SIRT1 pathway. What follows is an illustration of THC-mediated antioxidative regulation via the nuclear factor erythroid 2-related factor 2 (Nrf2) pathway, hypothesized to be involved in ocular diseases due to its regulation of multiple antioxidant enzymes [80]. Nrf2 physically interacts with Keap1, a negative regulator that limits Nrf2 activity. Under oxidative stress, modified Keap1 releases Nrf2, causing it to bind antioxidant response elements (AREs) in the nucleus. After that, the activation of AREs leads to the transcription of cytoprotective genes, including heme oxygenase-1 (HO-1), against oxidative stress [81]. Nrf2 deficiency rendered cells of the RPE more susceptible to stress and increased damage to these cells. This stress in RPE cells involved increased drusen-like deposits, the accumulation of lipofuscin, choroidal neovascularization, and the sub-RPE deposition of inflammatory proteins [82,83,84]. A recent study demonstrated that THC and OHC activate the liver’s Keap1-Nrf2 pathway. THC and OHC can occupy the Nrf2-binding site of Keap1, disturbing the binding of Nrf2 and Keap1 and resulting in Nrf2 translocation into the nucleus. Therefore, THC and OHC enhance the activation of Nrf2-targeted genes, including GCLC, GCLM, NQO1, and HO-1, against oxidative stress [85].

#### 3.3.1. THC has an Anti-Inflammatory Effect

Retinal ischemia is a common cause of vitreous neovascularization in retinal diseases, among which retinal vein occlusion and DR are characterized by retinal ischemia. Vitreous neovascularization is closely associated with local inflammation in the ischemic retina [86]. PGE_2_, one of the most important inflammatory mediators, is synthesized by COX-2. COX-2, however, is usually promoted by essential cytokines, including tumor necrosis factor-alpha (TNF-α), interleukin-1 beta (IL-1β), and interleukin-6 (IL-6), in the immune response to pathogenesis. PGE_2_ is a crucial factor in inflammatory diseases, fever, and pain [87]. Therefore, drugs to address various inflammatory diseases can be designed to target the proinflammatory cytokines COX-2 and PGE_2_. The most commonly adopted agents for inflammation in clinical practice are nonsteroidal anti-inflammatory drugs (NSAIDs) and COX-2 inhibitors; nevertheless, some NSAIDs can inhibit COX-1, which causes serious side effects such as gastrointestinal bleeding and ulcers [88]. Hence, it would be extremely desirable to explore selective COX-2 inhibitors as safe and efficient therapeutic agents for inflammatory conditions. Zhang et al. were the first to explore the pathways by which THC and OHC treatment exert an anti-inflammatory effect [25]. Their findings showed that THC and OHC suppressed the levels of TNF-α, IL-1β, and IL-6, demonstrating that THC and OHC could lessen inflammation by reducing the production of proinflammatory mediators. Moreover, the expression of COX-2 and PGE_2_ in tissues was eliminated by both THC and OHC in the study, while the expression of COX-1 remained unaffected.

#### 3.3.2. The Anti-VEGF Effect of THC

As noted in the previous section, VEGF has recently become a therapeutic target for eye diseases, especially AMD. Choroidal ischemia is one of the causes of AMD. VEGF is the primary cytokine related to angiogenesis, and ischemia induces VEGF expression through hypoxia-inducible factor-1α (HIF-1α) [89,90]. VEGF not only promotes cell proliferation but also increases vascular permeability through alterations in the phosphorylation of tight junction-related proteins (e.g., zonula occludens protein 1 (ZO1)) [91]. Additionally, VEGF triggers the MAPK signaling pathway, which is responsible for the proliferation of endothelial cells. Furthermore, VEGF-A, a member of VEGF, leads to upregulation of MMPs, which causes degradation of the matrix and increases the permeability of blood vessels [92]. Failure of the blood-retinal barrier (BRB) between RPE cells and retinal capillary endothelial (RCE) cells leads to disorders in the retina. Claesson-Welsh et al. indicated that long-term exposure to high glucose levels is associated with elevated expression of VEGF and induction of vascular permeability. Hence, age-related macular degeneration, inflammation, ischemia, and upregulation of VEGF are highly correlated with retinal diseases, mainly due to vascular permeability changes [93].

The resulting hypoxic environment aggravates VEGF. Therefore, proangiogenic stimulation of VEGF and VEGFR is a therapeutic target. A study conducted by Yoysungnoen et al. shed light on THC and VEGF mechanisms in cervical cancer. Significant reductions in HIF-1*α*, VEGF, and VEGFR-2 protein expression, as well as decreased microvascular density, were observed in a cervical cancer-implanted nude mouse model after THC administration [94]. In short, THC dramatically inhibited angiogenesis by downregulating HIF-1*α* and the VEGF/VEGFR-2 pathway.

#### 3.3.3. The Neuroprotective Effect of THC

The optic nerves are located at the back of the eyes and are injured in people with glaucoma. Gao et al. focused on THC’s potential use as a therapeutic agent for traumatic brain injury in rats. In their study, the expression levels of microtubule-associated protein 1A/1B-light chain 3 (LC3) and Beclin-1 were increased, and those of the ubiquitin-binding protein p62 were significantly decreased after THC treatment, indicating that THC modulated activation of the autophagy pathway, which was shown to play a protective role against brain trauma in rats [95]. Another study by Tyagi et al. illustrated the protective effects of THC associated with its antiautophagic effects [96]. However, in this case, THC’s administration was discovered to block the conversion of LC3-I to LC3-II. Therefore, THC inhibits the autophagy pathway and serves as a neuroprotective factor. Although it often acts as a double-edged sword, autophagy is an important self-protective mechanism in cells. Excessive autophagy can damage cells, but moderate autophagy may aid in neuronal survival because increased autophagic flux boosts the clearance of unnecessary proteins and damaged mitochondria [97]. The role of THC is to balance excess and deficient autophagy. In terms of autophagy in the eye, autophagy inhibition is a promising target for preventing retinal ganglion cell degeneration and axonal degeneration in glaucoma [97,98]. In summary, the modulation of autophagy through THC administration may be an important neuroprotective intervention in glaucomatous neuropathies.

#### 3.3.4. The Inhibitory Effect of THC on Platelet Aggregation

Several coagulation factors are linked to proliferative DR. The β-thromboglobulin concentration was higher, the platelet factor 4 level was significantly increased, and fibrinogen was found to be aggregated in patients with DR compared with controls [99]. β-thromboglobulin and platelet factor 4 are two proteins involved in platelet activation, and fibrinogen, an immediate precursor of fibrin, induces platelet aggregation through the COX pathway.

A study comparing the effects of THC and curcuminoids on human platelet aggregation and blood coagulation was conducted by Chapman et al. The results showed that all curcuminoids, with the exclusion of curcumin, reduced platelet aggregation and that THC was the most potent curcuminoid. THC and other curcuminoids were found to act by inhibiting the ability of the COX enzyme to synthesize the formation of proinflammatory thromboxanes [69]. In addition, the effect of curcuminoids in reversing aggregation is mostly due to platelet aggregation induced by arachidonic acid [100]. However, the pathways involved in platelet aggregation are strongly related to different factors, and these experimental antiplatelet effects have not yet been confirmed in vivo due to the low bioavailability of current curcumin derivatives. Therefore, although new formulations of curcumin and THC may hold therapeutic promise, future studies are required to understand the antiplatelet effects of these compounds on the eye (Figure 4).

## 4. Conclusions

Curcumin has been used throughout history for the prevention of various conditions. The potential benefits of curcumin in several major ocular diseases, such as age-related cataract, glaucoma, AMD, and DR, are under investigation. However, the low bioavailability of curcumin limits its effective concentration. Therefore, two main approaches to overcome this issue were discussed above: the formulation of curcumin for its delivery and the use of curcumin metabolites. The former has long been intensively studied and can roughly be summarized as three strategies: lipid addition, absorption and dispersion on matrices; particle size reduction; and surface property modulation. Pharmaceutical applications of cumin usually combine several methods to reach its ideal application. The latter is a relatively new field to researchers and pharmaceutical companies, and among curcumin metabolites and conjugates, THC has been the focus. Ocular diseases lead to visual impairments through oxidative stress, ER stress, inflammation, and autophagy. However, THC was found to possess antioxidative, anti-inflammatory, anti-VEGF, and neuroprotective properties in vivo and in vitro.

Combined with the above arguments, this review suggests the potential effect of THC against ocular impairment. However, few direct experiments of THC have been conducted in the eyes.

First, since vessels in the eye belong to the peripheral vascular system, achieving a therapeutic level of curcumin in the eye is difficult. Once the limitation of curcumin bioavailability has been overcome, ensuring the stability of curcumin in other organs, such as the liver, spleen, neural system, and cardiovascular system, is an additional issue. Fortunately, THC was reported to have better bioavailability and stability than curcumin, explaining the attention on THC in recent years. In addition, botanical compounds are generally used as prophylactic treatments instead of remedies. In general, further investigation is required for curcumin and its related compounds to be applied as noninvasive and preventative complementary compounds against eye diseases.

## Figures and Tables

**Figure 1 ijms-22-00212-f001:**
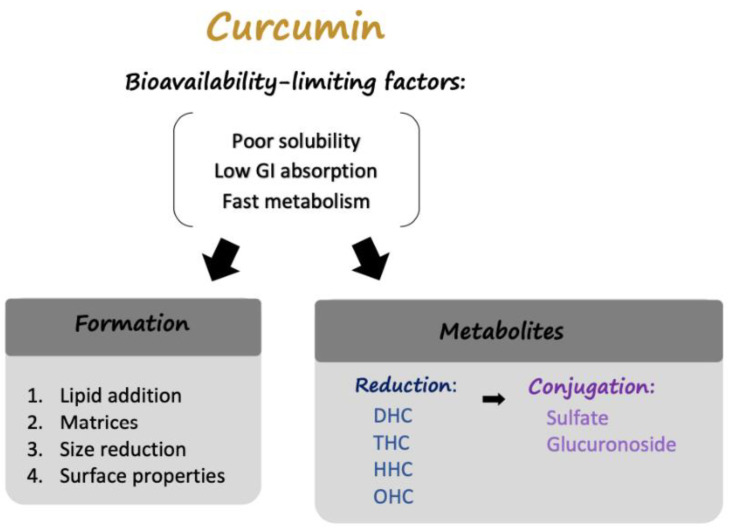
The factors limiting curcumin bioavailability and solutions. Poor solubility, low gastrointestinal absorption, and fast hepatic and intestinal metabolism are the main factors that constrain curcumin utilization by the human body. Studies on the delivery of curcumin in different formulations and metabolites of curcumin have been conducted to address these factors. Four strategies to address curcumin delivery are summarized in this review: lipid addition, absorption and dispersion on matrices, size reduction, and alterations to surface properties. Regarding the metabolites of curcumin, there are several reduced forms of curcumin, each of which can be conjugated to a sulfate group or glucuronic acid.

**Figure 2 ijms-22-00212-f002:**
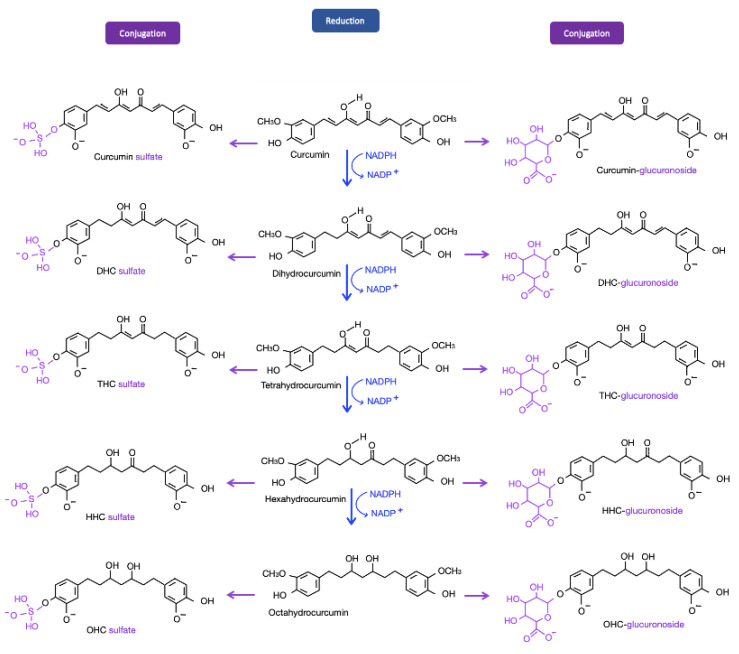
Metabolites of curcumin. After being taken up by enterocytes, curcumin molecules are metabolized. Through reductase, sulfotransferase, and glucuronosyltransferase enzymes, the systemic level of curcumin is reduced. Several metabolites of curcumin are shown; among these metabolites, tetrahydrocurcumin (THC) has been the focus of recent studies and is discussed in later paragraphs. Pathways in blue represent the reductions of curcumin by NADPH; pathways in purple are the conjugations of curcumin and its following reductions.

**Figure 3 ijms-22-00212-f003:**
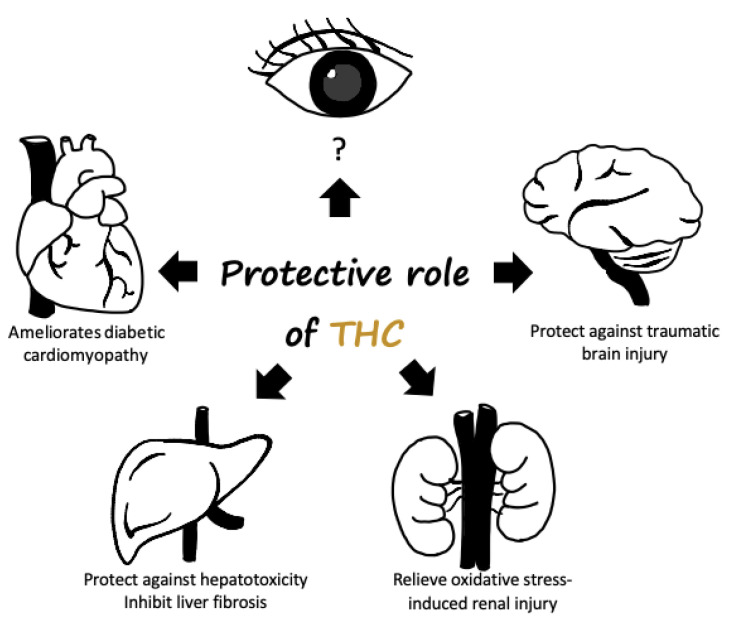
Potential protective role of tetrahydrocurcumin in the human body. THC administration may ameliorate diabetic cardiomyopathy and renal damage. In addition, THC is observed to possess superior hepatic-protective effects and the antiautophagic effect was shown to participate in brain trauma after THC administration.

**Figure 4 ijms-22-00212-f004:**
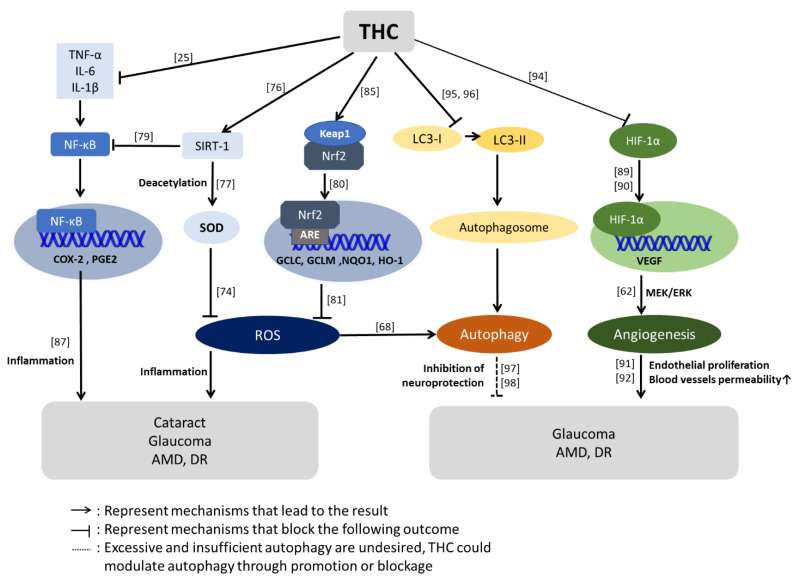
Possible pathways of THC-mediated protective effect on eye diseases. Regarding antioxidative stress pathways, the activation of sirtuin-1 (SIRT-1) was increased after THC treatment. SIRT-1 deacetylates SOD-1 and SOD-2 are ROS (reactive oxygen species) scavengers. However, SIRT-1 also inhibits nuclear factor-κB (NF-κB), leading to the alleviation of inflammation and oxidative stress. In addition to the SIRT-1 pathways, THC can activate antioxidative stress genes, such as heme oxygenase-1 (HO-1), by occupying the Nrf2-binding site of Keap1. Regarding the modulation of autophagy, THC balances excess and deficient autophagy. About the effect of THC against angiogenesis, THC downregulates the expression of hypoxia-inducible factor-1α (HIF-1*α*) and vascular endothelial growth factor (VEGF). As noted, because of the collective antioxidative, anti-inflammatory, and antiautophagic effects of THC and its ability to inhibit VEGF, THC plays a neuroprotective role. These THC pathways were observed in different experiments but may have valuable benefits in protecting the eye. Abbreviations: SOD: superoxide dismutase, Nrf2: nuclear factor erythroid 2-related factor 2. The different numbers in the figures are reference citations.

**Table 1 ijms-22-00212-t001:** Sites, pathogenic mechanisms, and main treatment strategies for the four ophthalmic conditions discussed in this review.

	Site	Pathogenesis	Treatment	
Age-related cataract	Lens	Malfunctioning proteins on lens	(Present) removal of the cloudy lens (Future) antioxidant application	[44]
Glaucoma	Optic nerves	Ganglion cell degeneration	(Present) release of IOP (Future) neuroprotection	[54]
AMD	Retina (macula)	Macular degeneration and vascularization	(Future) anti-VEGF agents and reversal of angiogenesis	[57]
DR	Retina	Vascular abnormalities	(Traditional) laser or surgery (Future) anti-VEGF agents	[70]

IOP, intraocular pressure.

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
