# Peer review of "Curcumin Metabolite Tetrahydrocurcumin in the Treatment of Eye Diseases"

_ijms, 2020, doi:10.3390/ijms22010212_

Round 1

Reviewer 1 Report

The authors have successfully addressed each of my comments.

Reviewer 2 Report

None

This manuscript is a resubmission of an earlier submission. The following is a list of the peer review reports and author responses from that submission.

Round 1

Reviewer 1 Report

This is a review article focusing on the potential role of curcumin’ metabolites as a pharmacological strategy. In the last years, curcumin and its metabolites have become an interesting public health issue. Interestingly, recent studies suggested that tetrahydrocurcumin (THC) metabolite could be useful in ocular pathologies.

Line 59:The authors reported that increasing age represents a worrying cause of visual impairment and blindness worldwide. Please, the authors should specify if there are other risk factors involved in this public health phenomenon. 

Line 67:The authors declared that curcumin possesses diverse properties. Furthermore, several studies demonstrated that curcumin could be considered a wound healing agent. Please consider this paper DOI: 10.3390/ijms20051119

Line 79-81.  The authors cited different delivery systems for curcumin. I think that would be really helpful to include a brief description of each system’ characteristics, their mechanisms action and their derivation.

Line 257-272: In this paragraph the authors discuss on the diabetic retinopathy. Specifically, they declared that VEGF, oxidative stress, ER stress, inflammation, and autophagy represent the most common pathogenic mechanisms involved. Could the authors provide some details on this topic, also considering this paper DOI:10.1155/2018/5407482

Line 273: Table 1 summarizes sites, pathogenic mechanisms, and main treatment strategies involved in the ophthalmic conditions considered.  I think that would be really helpful to make the table more effective and easier to interpret.

Line 293:The authors declare that ROS play a crucial role in eye pathogenesis, focusing on the Superoxide dismutase (SODs) only. Could the authors discuss on this topic, also considering other ROS?

Reviewer 2 Report

In this review of curcumin and curcuminoid pharmacology in the eye the authors discuss a spectrum of diseases affecting various ocular tissues. While the coverage of curcumin action appears fairly complete, the review suffers from inaccuracies regarding the various eye tissues and their pathologies.  Specifiic comments follow:
1. 1, Introduction, general: As curcumin has a long history as a remedy and preventive, a very brief description of the historical use of curcumin both in Western and also Ayurvedic medicine would be helpful and informative here.
2. 2.2.4, Surface..., p. 3, lines 129-130, "Furthermore, the administration of Cur-BSA-NPs-Gel (curcumin incorporated into albumin nanoparticles) to rabbits in an ophthalmic experiment showed that Cur-BSA-NPs-Gel is considered safe for ophthalmic drug delivery." These are animal experiments and can suggest but not prove or show drug safety in humans.
3. 2.2.4, Surface..., p. 3, lines 132-137, "Finally, the role of curcumin-encapsulated MePEG-PCL nanoparticles (methoxypoly(ethylene glycol)-poly(caprolactone)) in the prevention of corneal neovascularization was successfully characterized.": Once more, these are studies in rats. Please provide the model system for these studies as the results can vary with species.
4. 2.3, Modulation..., general, and Fig. 2: It would be helpful to have the actual enzyme names (rather than the activity) either in the text, the figure, or the figure legend. For example, are all reduction reactions carried out by a single reductase, or is there a reductase specific for each compound?
5. 3.1., Main ophthalmic conditions in modern society, general: It might be helpful to mention that most of these diseases are increasing in various populations, some dramatically, related both to age shifts and increasing risk factors.
6. 3.1.1., Cataract, lines 198-200, "The main pathogenesis of cataract is the modification of lens proteins under oxidative stress. Modified proteins fail to carry out 199 their metabolic activity, causing inflammation and necrosis.": Age related cataract, but not congenital or childhood cataracts are probably related to oxidative and uv damage. The authors should be careful to make that distinction, perhaps renaming the section. In addition, most studies show damage to the lens crystallins rather than enzymes, and the damage to them is more structural than functional. The authors might wish to discuss the chaperone activity of alpha-crystallins in this process, especially as curcumin has been show to protect this under oxidative stress. Finally, inflamation has not been shown to play a large role in cataractogenesis.
7. 3.1.1, Cataract, p. 6, lines 200-210, "A cataract is initiated when free radicals damage the eye lens by causing lipid peroxidation (LPO) and the aggregation of malfunctioning proteins. [29] Babizhayev, Deyev, and Linberg injected LPO products into the vitreous after finding a correlation between the accumulation of a fluorescent end product of LPO and the degree of lens opacity. The injection of LPO products induced cataract, implying that peroxide-induced damage of the lens fiber may initiate cataractogenesis. [30]": This is a bit misleading, as there are many contributors to age related cataract in addition to free radicals, and lipid peroxidation is only one of many results of oxidative stress.
7. 3.1.1 Cataract, p. 6, lines 208-210, "Redox reactions between SH-containing proteins and glutathione (GSH) result in the accumulation of malfunctioning proteins and a decrease in reduced GSH; therefore, GSH accelerates cataract formation. [31] [29, 32]": Actually GSH is one of the main buffers of the lens against oxidative damage. This needs to be revised.
8.  3.1.1 Cataract, p. 6, "An increased level of αB-crystallin unfolding in response to stress is found in cataract and leads to endoplasmic reticulum (ER) stress in the lens.": Alpha-crystallins are actually among the last lens proteins to unfold, and protect the other crystallins from aggregation. ER stress generally occurrsly after this protective system fails.
9. 3.1.2. Glaucoma, p. 6, lines 228-230, "Other factors include chemical injury, inflammatory conditions, and blood vessel blockage. [35]": Reference 35 only discusses increased pressure and glutamate toxicity from ischemic damage.
10. 3.1.3. AMD, p. 7, lines 243-245, "Several pathways, such as lipofuscin accumulation in RPE, choroidal ischemia, and oxidative damage, have been implicated in the pathogenesis of AMD.": Lipofuscin accumulation is not itself a pathway, but rather a consequence of oxidative damage.
11. 3.1.4. DR, p. 7, lines 262-265, "A number of pathogenic mechanisms are involved in DR, including, for example, VEGF, oxidative stress, ER stress, inflammation, and autophagy. Hyperglycemia leads to dysfunction of the electron transport chain, thus causing the accumulation of reactive oxygen species (ROS) in mitochondria.": The best data indicate that DR occurrs through chronically elevated blood glucose elevating PKC and MAPK, inhibiting PDGFd, resulting in apoptosis of pericytes.
12. 3.3. Effect of THC on antioxidative stress, p. 9, lines 292-294, "The eye is constantly exposed to all kinds of oxidative stress, including ionizing radiation and photooxidation (ultraviolet light).": Except for therapeutic radiation, there is little support for ionizing radiation in occular disease, and UV light does not penetrate beyond the lens.
13. 3.3. Effect of THC on antioxidative stress, p. 9, lines 300-306, "SOD-1 knockout in mice was observed to increase the risk of cataract and macular degeneration development. [50]": Reference 50 does not mention cataracts. Also, reference 52 relates to diabetic cardiomyopathy. Similarly, reference 56 does not mention cataract.
14. 3.3.2. The anti-VEGF effect of THC, p. 10, lines 363-364, "The leakage of blood contributes to increased edema in the eye, making it more difficult for vessels to provide efficient blood flow.": Vascular permeability here does not relate to extravasation of blood, but rather small solutes.
15. General: While the manuscript is fairly intelligible, there are numerous problems with English usage and grammar that make it difficult to follow in places.  It would benefit from careful reading by a line editor.